# Differentially Private Image Classification from Features

**Harsh Mehta**                                                    *harshm@google.com*
*Google Research*

**Walid Krichene**                                                 *walidk@google.com*
*Google Research*

**Abhradeep Thakurta**                                           *athakurta@google.com*
*Google Research*

**Alexey Kurakin**                                                 *kurakin@google.com*
*Google Research*

**Ashok Cutkosky**                                               *ashok@cutkosky.com*
*Boston University*

**Reviewed on OpenReview:** *https://openreview.net/forum?id=Cj6pLclmwT*

## Abstract

In deep learning, leveraging transfer learning has recently been shown to be an effective strategy for training large high performance models with Differential Privacy (DP). Moreover, somewhat surprisingly, recent works have found that privately training just the last layer of a pre-trained model provides the best utility with DP. While past studies largely rely on using first-order differentially private training algorithms like DP-SGD for training large models, in the specific case of privately learning from features, we observe that computational burden is often low enough to allow for more sophisticated optimization schemes, including second-order methods. To that end, we systematically explore the effect of design parameters such as loss function and optimization algorithm. We find that, while commonly used logistic regression performs better than linear regression in the non-private setting, the situation is reversed in the private setting. We find that least-squares linear regression is much more effective than logistic regression from both privacy and computational standpoint, especially at stricter epsilon values ($\varepsilon < 1$). On the optimization side, we also explore using Newton's method, and find that second-order information is quite helpful even with privacy, although the benefit significantly diminishes with stricter privacy guarantees. While both methods use second-order information, least squares is more effective at lower epsilon values while Newton's method is more effective at larger epsilon values. To combine the benefits of both methods, we propose a novel optimization algorithm called DP-FC, which leverages feature covariance instead of the Hessian of the logistic regression loss and performs well across all $\varepsilon$ values we tried. With this, we obtain new SOTA results on ImageNet-1k, CIFAR-100 and CIFAR-10 across all values of $\varepsilon$ typically considered. Most remarkably, on ImageNet-1K, we obtain top-1 accuracy of 88% under DP guarantee of $(8, 8 * 10^{-7})$ and 84.3% under $(0.1, 8 * 10^{-7})$.

## 1 Introduction

Despite impressive performance, large machine learning models are susceptible to attacks. Previous work has demonstrated successful membership attacks where the goal is to extract exact instances of training

---

Code: *https://github.com/google-research/google-research/tree/master/dp_transfer*

examples that the model was trained on (Shokri et al., 2017; Carlini et al., 2019; 2021; Choquette-Choo et al., 2020; Liu et al., 2021b; Balle et al., 2022) and models of larger size are known to be more likely to memorize training data. These attacks can be quite egregious if the model was trained on sensitive data like personal photos or emails. One approach to mitigate this risk is to train such models with privacy guarantees. In particular, Differential Privacy (DP) has become the gold standard in quantifying the risk and providing formal guarantees against membership attacks (Nasr et al., 2021).

Informally, Differentially Privacy implies that an adversary learns almost the same thing about an individual data point independent of their presence or absence in the training data set. More formally, DP is defined as follows. Two datasets $D, D'$ are called neighboring datasets if one is obtained from the other by deleting one data point.

**Definition 1.1** (Differential Privacy (Dwork et al., 2006b;a))**.** *A randomized algorithm $\mathcal{A}$ is $(\varepsilon, \delta)$-differentially private if, for any pair of neighboring datasets $D$ and $D'$, and for all events $\mathcal{S}$ in the output range of $\mathcal{A}$, we have*

$$\mathbf{Pr}[\mathcal{A}(D) \in \mathcal{S}] \le e^{\varepsilon} \cdot \mathbf{Pr}[\mathcal{A}(D') \in \mathcal{S}] + \delta,$$

*where the probability is over the randomness of $\mathcal{A}$.*

In the field of deep learning, Differentially Private Stochastic Gradient Descent (DP-SGD) (Song et al., 2013; Bassily et al., 2014; Abadi et al., 2016) is the most commonly used method for training models with DP guarantees. While DP-SGD is a fairly general algorithm, a naive application can suffer from several computational challenges. To make matters worse, the gap between performance of a model with and without privacy typically widens as the model is made larger. This stands as a significant obstacle to a wider adoption and deployment of large practical models with privacy guarantees.

For the problem of Image Classification, several works have shown that transfer learning can be a very effective strategy in order to improve privacy-utility trade-off when formal privacy guarantees are required (Kurakin et al., 2022; Mehta et al., 2022; De et al., 2022). In this setting, a model is first pre-trained with "non-sensitive" data without privacy guarantees, then fine-tuned on a "sensitive" dataset over which a formal privacy guarantee is required. Similar to previous works, we simulate several publicly available image classification benchmarks (like ImageNet) as "sensitive" datasets.

Interestingly, Mehta et al. (2022); De et al. (2022) observe that privately fine-tuning just the last layer of a pre-trained model (using DP-SGD) leads to state of the art results in the ImageNet-1k dataset. This is quite fortuitous, since privately fine-tuning the full model typically introduces significant computational challenges. We build on this observation, and perform a comprehensive exploration of various design parameters, including the choice of loss function and optimization algorithm, beyond simple DP-SGD. In this restricted setting of learning a single layer privately using features extracted from a pre-trained model, more sophisticated methods, such as second-order methods, are computationally viable. Our main contributions are as follows:

- Somewhat surprisingly, we find that linear regression solved using DP Least Squares performs much better than logistic regression solved using DP-SGD, especially at lower epsilons.

- Postulating that the benefits largely stem from the use of second-order information in the least squares solution, we further explore using Newton's method to solve logistic regression. While Newton's method outperforms linear regression in the non-private setting, we find that it still performs worse with privacy constraints, largely because sanitizing the Hessian with logistic regression requires adding far more noise than in linear regression, where part of the Hessian can be shared across all classes.

- To combine the benefits of both, we introduce a method which we call Differentially Private SGD with Feature Covariance (abbreviated as DP-FC) where we simply replace the Hessian in Newton's method with sanitized Feature Covariance. Using Feature Covariance instead of Hessian allows us to make use of second-order information in the training procedure while sharing it across classes and iterations, which greatly reduces the amount of noise that needs be added to sanitize it. This allows us to continue using logistic regression, which performs better in non-private setting, while benefiting from improved privacy-utility trade-off as seen with linear regression in the private setting.

- With DP-FC, we surpass previous state of the art results considerably on 3 image classification benchmarks, namely ImageNet-1k, CIFAR-10 and CIFAR-100, just by performing DP fine-tuning on features extracted from a pre-trained model, see Table 1 for a summary. Consistent with previous works, we also find that performance increases as the pre-training dataset and the model are made larger.

| Dataset | Epsilon | Previous SOTA | Accuracy | Method | Epochs (= Steps) | Pretraining DS |
|---|---|---|---|---|---|---|
| CIFAR-10 | 0.01 | | 95.55 (0.98) | DP-LS | 1 | JFT |
| | 0.05 | | 97.81 (0.22) | DP-FC | 10 | JFT |
| | 0.1 | | 98.23 (0.12) | DP-FC | 10 | JFT |
| | 0.5 | | 98.73 (0.04) | DP-FC | 10 | JFT |
| | 1.0 | 96.7 | 98.80 (0.05) | DP-FC | 10 | JFT |
| | 2.0 | 97.1 | 98.80 (0.03) | DP-FC | 10 | JFT |
| | 4.0 | 97.2 | 98.83 (0.02) | DP-FC | 10 | JFT |
| | 8.0 | 97.4 | 98.84 (0.02) | DP-FC | 10 | JFT |
| | $\infty$ | | 98.90 (0.00) | LS | 1 | JFT |
| CIFAR-100 | 0.01 | | 76.96 (0.08) | DP-LS | 1 | I21K |
| | 0.05 | | 78.31 (0.33) | DP-LS | 1 | JFT |
| | 0.1 | | 80.57 (0.26) | DP-LS | 1 | JFT |
| | 0.5 | | 84.82 (0.38) | DP-FC | 10 | JFT |
| | 1.0 | 83.0 | 87.85 (0.14) | DP-FC | 10 | JFT |
| | 2.0 | 86.2 | 88.77 (0.16) | DP-FC | 10 | JFT |
| | 4.0 | 87.7 | 89.51 (0.17) | DP-FC | 10 | JFT |
| | 8.0 | 88.4 | 89.78 (0.08) | DP-FC | 10 | JFT |
| | $\infty$ | | 90.60 (0.00) | LS | 1 | JFT |
| ImageNet-1K | 0.01 | | 81.99 (0.08) | DP-LS | 1 | JFT |
| | 0.05 | | 83.74 (0.10) | DP-LS | 1 | JFT |
| | 0.1 | | 84.28 (0.06) | DP-FC | 1 | JFT |
| | 0.5 | | 86.04 (0.06) | DP-FC | 10 | JFT |
| | 1.0 | 84.4 | 86.78 (0.07) | DP-FC | 10 | JFT |
| | 2.0 | 85.6 | 87.34 (0.04) | DP-FC | 10 | JFT |
| | 4.0 | 86.0 | 87.70 (0.03) | DP-FC | 10 | JFT |
| | 8.0 | 86.7 | 88.02 (0.03) | DP-FC | 10 | JFT |
| | $\infty$ | | 88.90 (0.00) | Newton | 10 | JFT |

Table 1: Compilation of our best private Top-1 test accuracies. We report median and standard deviation across 5 training runs with different seeds. All number are SOTA across all epsilons to the best of our knowledge. Previous state of the art for CIFAR-10 and CIFAR-100 were reported form Bu et al. (2022a) and for ImageNet-1K from De et al. (2022). We denote $\varepsilon$ to be $\infty$ for non-private setting where we turn off all sanitization steps including clipping. We set $\delta$ to $8 * 10^{-7}$ for ImageNet-1k, and $1 * 10^{-5}$ for CIFAR-10 and CIFAR-100. Interestingly, in the non-private setting, most previous works (including Zhai et al. (2021)) use Linear Regression when finetuning from features but we found that Logistic Regression (solved using Newton's method) performs much better and leads to an impressive 88.9% accuracy when finetuning just the last layer. To put this in perspective, this is only 1.1% less than the current state of the art non-private accuracy of 91% on ImageNet-1k (Yu et al., 2022b). We report extensive hyperparameter details in the appendix for reproducibility of our results.

## 2 Private Learning from Features

In this section, we describe the details of optimization strategies we considered, and state the privacy guarantees for each of them.

Given a data set $\mathcal{D} = \{(x_1, y_1), \cdots, (x_n, y_n)\}$, we optimize the function $\mathcal{L} : \mathbb{R}^{m \times d} \to \mathbb{R}$ defined as follows

$$\mathcal{L}(\theta) \triangleq \frac{1}{n} \sum_{i=1}^{n} \sum_{j=1}^{m} \ell(\langle \theta_j, x_i \rangle, y_{ij}) \tag{1}$$

where $n$ is the number of examples, $m$ is the number of classes, $\theta \in \mathbb{R}^{m \times d}$ is the weight matrix to be learned, $x_i$ is the feature vector for example $i$, $y_{ij}$ is the label of example $i$ and class $j$, and $\ell$ is a convex loss function. We assume that $y_{ij} \in [0, 1]$ for all $i, j$. Additionally, we also use the short hand $\ell(\theta; (x_i, y_i)) = \sum_{j=1}^{m} \ell(\langle \theta_j, x_i \rangle, y_{ij})$ where helpful in order to simplify the notation.

In the case of learning from features extracted from a pre-trained model, the feature vectors $x_i$ are last layer features. Further, unless otherwise specified, we will assume $\ell$ to be the logistic loss, i.e. $\ell_{\text{logistic}}(z, y) = -y \log \sigma(z) - (1 - y) \log(1 - \sigma(z))$ where $\sigma(z) = \frac{1}{1+e^{-z}}$. Finally, we will also assume that each step of optimization considers the whole batch, which greatly simplifies both the privacy analysis of algorithms and the experiments. Rest of this section includes descriptions of several iterative solvers of this minimization problem, both in non-private and private settings. Some of these methods rely on the fact that we are only interested in fine-tuning just the last layer, while others are more general. In the privacy analysis, we use zCDP (zero - Concentrated Differential Privacy) (Bun & Steinke, 2016), but we state our empirical results always with final privacy guarantee in $(\varepsilon, \delta)$-DP terms, as done in previous works.

## 2.1 DP-SGD

Arguably the most popular approach to solving the above minimization problem in the non-private setting is Stochastic Gradient Descent (SGD). In the full-batch setting, at every iteration, SGD performs the update:

$$g_t \leftarrow \frac{1}{n} \sum_{i=1}^{n} \nabla \ell(\theta_t; (x_i, y_i)) \qquad \theta_{t+1} = \theta_t - \eta_t g_t \tag{2}$$

where $\eta_t$ denotes the learning rate used for for iteration $t$.

DP-SGD is a private variant of this algorithm and the baseline in all our experiments. Computationally, in order to bound the sensitivity of each training example, Abadi et al. (2016) suggest computing a gradient for each example separately and clipping each to a maximum norm of $C$ (a user-specified hyper-parameter):

$$g_t \leftarrow \sum_{i=1}^{n} \text{clip}\left(\nabla \ell(\theta_t; (x_i, y_i))\right) \qquad \tilde{g}_t \leftarrow \frac{g_t + \mathcal{N}\left(0, (\sigma C)^2\right)}{n} \tag{3}$$

where $\text{clip}(v) = v \cdot \min\left\{1, \frac{C}{\|v\|_2}\right\}$.

After summing the clipped example gradients, a noise vector sampled from a Gaussian distribution with standard deviation $\sigma C$ is added, where $\sigma$ is a parameter that determines the privacy guarantee via the Gaussian mechanism.

As shown in Algorithm 4, once the gradient has been sanitized, we are free to use it to accumulate statistics (e.g. first or second moment estimates) which are typically useful with the optimization process. Algorithm 4 presents a generalized version of DP-SGD where the gradients get processed in the traditional DP-SGD fashion, and are then passed to a first-order optimizer as an input. This lets us instantiate DP versions of well-known first-order optimizers like SGD, Momentum and Adam. We employ DP-Adam in all our experiments. Finally, we omit the privacy analysis for the DP-SGD baseline since it is standard, but we do include in the appendix the details of the implementation we used for translation from privacy constraints to noise scale.

## 2.2 DP-Newton

Since the optimization problem under consideration is a relatively simple convex problem, second-order DP algorithms can be viable. We first consider a privatized version of the popular Newton's method and denote

it as DP-Newton. To control the sensitivity of each training example, one naive approach is to compute per-example Hessians and clip their norm, in a similar fashion to example gradient clipping in DP-SGD. But even in our last-layer fine-tuning setting, this can be prohibitively expensive. For instance, training on features extracted from ViT-G for ImageNet-1k fine-tuning, Hessian tensor (in a block diagonal form) is of size [1000, 1664, 1664] with approximately 2.8B entries. In order to avoid instantiating the Hessian for every training example, we instead choose to clip the feature vectors $x_i$, then translate the clipping threshold into bounds on the Hessian and gradient norms. This is summarized in Algorithm 1.

---

**Algorithm 1** Differentially Private Newton's Method

---

**Require:** Data set $D = \{(x_1, y_1), \cdots, (x_n, y_n)\}$ with $(x_i, y_i) \in \mathcal{D}$, loss function: $\ell : \mathbb{R}^d \times \mathbb{R} \to \mathbb{R}$, regularization coefficient $\lambda$, learning rate $\eta$, clipping norm: $C$, number of iterations: $T$, noise multiplier: $\sigma$, bound on the second derivative of the loss: $\beta_H$

1: Clip all features: $\tilde{x}_i \leftarrow \mathsf{clip}(x_i)$ for all $i \in \{1, \ldots, n\}$ where $\mathsf{clip}(v) = v \cdot \min\left\{1, \frac{C}{\|v\|_2}\right\}$.

2: Randomly initialize $\theta_0$.

3: **for** $t = 1, \ldots, T$ **do**

4:     **for** $j = 1, \ldots, m$ **do**

5:         $g_{t,j} \leftarrow \sum_{i=1}^n \nabla\ell(\theta_{t,j}^\top \tilde{x}_i, y_{ij})$.

6:         $H_{t,j} \leftarrow \sum_{i=1}^n \nabla^2\ell(\theta_{t,j}^\top \tilde{x}_i, y_{ij}) + \lambda I$

7:         $\tilde{g}_{t,j} \leftarrow \frac{1}{n}g_{t,j} + \mathcal{N}\left(\vec{0}_d, \left(\frac{\sigma C\sqrt{m}}{n}\right)^2\right)$ where $\mathcal{N}\left(\vec{0}_d, \left(\frac{\sigma C\sqrt{m}}{n}\right)^2\right)$ indicates a $d$-dimensional vector each of whose coordinates is an i.i.d. Gaussian with standard deviation $\sigma C\sqrt{m}/n$.

8:         $\tilde{H}_{t,j} \leftarrow \frac{1}{n}H_{t,j} + \mathcal{N}\left(\vec{0}_{d\times d}, \left(\frac{\sigma \beta_H C^2\sqrt{m}}{n}\right)^2\right)$ where $\mathcal{N}\left(\vec{0}_{d\times d}, \left(\frac{\sigma \beta_H C^2\sqrt{m}}{n}\right)^2\right)$ indicates a $d \times d$ matrix each of whose coordinates is an i.i.d. Gaussian with standard deviation $\sigma\beta_H C^2\sqrt{m}/n$.

9:         $\theta_{t+1,j} \leftarrow \theta_{t,j} - \eta\tilde{H}_{t,j}^{-1}\tilde{g}_{t,j}$

10:     **end for**

11: **end for**

12: **return** $\theta_T$.

---

**Theorem 2.1** (Privacy guarantee for Algorithm 1)**.** *Suppose $\ell$ is twice differentiable, and that for all $(z, y)$, $|\ell'(z, y)| \leq 1$ and $|\ell''(z, y)| \leq \beta_H$. Then, Algorithm 1 satisfies $\frac{T}{\sigma^2}$-zCDP.*

Here $\ell'$ and $\ell''$ denote the first and second derivatives of $\ell$ with respect to its first argument. We choose the bound on the first derivative to be 1 without loss of generality (via scaling of $\ell$). Note that the assumptions of the theorem are satisfied for the sigmoid cross-entropy loss (i.e. logistic regression) with $\beta_H = \frac{1}{4}$. Indeed, $|\ell'_{\text{logistic}}(z, y)| = |y - \sigma(z)|$, and $|\ell''_{\text{logistic}}(z, y)| = \sigma(z) * (1 - \sigma(z))$ where $\sigma(z) = \frac{1}{1+e^{-z}}$. The first expression is bounded by 1 (since $y, \sigma(z) \in [0, 1]$) and the second expression is bounded by $\frac{1}{4}$.

For the squared loss $\ell(z, y) = \frac{1}{2}(z - y)^2$, notice that (non-private) Newton's method reaches the solution in a single step, regardless of where it is initialized. Therefore it suffices to initialize Algorithm 1 at $\theta_0 = 0$, and run it for a single step ($T = 1$). When initialized at $\theta = 0$, we have $z = 0$ and the assumptions of the theorem are satisfied (first and second derivatives are bounded, with $\beta_H = 1$).

*Proof.* The crux of the proof is to ensure that Lines 7 and 8 individually satisfy $\frac{1}{2m\sigma^2}$-zCDP. The rest of the proof is just simple composition of zCDP Bun & Steinke (2016) across $m$ classes and $T$ iterations. To see this, first let $D$ and $D'$ be neighboring datasets and let $(x, y)$ be the differing datapoint between $D$ and $D'$.

Now, notice that in the computation of $g_t$, we have the following: $\|g_{t,j}(D) - g_{t,j}(D')\| = \|\ell'(\langle\theta_{t,j}, \tilde{x}\rangle, y_j)\tilde{x}\|$. Thus, since $\|\tilde{x}\|_2 \leq C$, and $|\ell'(\langle\theta_j, x\rangle, y_j)| \leq 1$:

$$\|g_{t,j}(D) - g_{t,j}(D')\|_2 \leq C \tag{4}$$

From equation (4) it immediately follows that the computation of $g_{t,j}$ for each $t \in \{0, \ldots, T-1\}$ and each $j \in \{1, \ldots, m\}$ satisfies $\frac{1}{2m\sigma^2}$-zCDP. Now, moving on to the sensitivity of $H_t$ in Line 8, we have

$H_{t,j}(D) - H_{t,j}(D') = \ell''(\langle \theta_j, \tilde{x} \rangle, y_j)\tilde{x}\tilde{x}^\top$. Then, since $\|\tilde{x}\|_2 \leq C$ and $|\ell''(\langle \theta_j, x \rangle, y_j)| \leq \beta_H$, we have:

$$\|H_{t,j}(D) - H_{t,j}(D')\|_F \leq \beta_H C^2, \tag{5}$$

and equation (5) immediately implies that the computation of $H_{t,j}$ for each $t \in \{0, \ldots, T-1\}$ and $j \in \{1, \ldots, m\}$ satisfies $\frac{1}{2m\sigma^2}$-zCDP. This completes the proof. $\qquad \square$

## 2.3 DP-LS

One drawback of DP-SGD and DP-Newton is that each training example $i$ contributes to the gradients (resp. Hessians) of all classes $j \in \{1, \ldots, m\}$, so the sensitivity (and hence the scale of required noise) increases with the number of classes. This is visible in Algorithm 1 where the amount of noise added for privacy protection scales with $\sqrt{m}$ (Lines 7 and 8). When the number of classes is large, this reduces signal-to-noise ratio and can hurt quality. In this section, we develop a method that aims to address this issue. We take inspiration from the matrix factorization literature, in which one can separate the loss function into the contribution of positive and negative classes. We assume that labels are binary ($y_{ij} \in \{0, 1\}$), and we consider the following quadratic loss function:

$$\mathcal{L}(\theta) = \frac{1}{2} \sum_{i=1}^{n} \sum_{j=1}^{m} y_{ij}(x_i^\top \theta_j - y_{ij})^2 + \frac{\alpha}{2}(x_i^\top \theta_j)^2 + \frac{\lambda}{2}\|\theta_j\|^2,$$

in other words, we take $\ell(z, y) = \frac{1}{2}(y(z-y)^2 + \alpha z^2)$. The first term fits the positive labels (notice that this term vanishes when $y = 0$), while the second term fits negative labels, and $\alpha$ is hyper-parameter that trades-off the two terms. This formulation was studied by Hu et al. (2008) and enjoys remarkable empirical success Koren & Bell (2015). It turns out that this method is also well-suited for privacy, as we shall discuss below.

By expanding the quadratic terms, we can write the loss as

$$\mathcal{L}(\theta) = \frac{1}{2} \sum_{j=1}^{m} \left( \theta_j^\top A_j \theta_j - 2\theta_j^\top b_j + \alpha \theta_j^\top G \theta_j + \lambda \theta_j^\top \theta_j \right)$$

where for all $j \in \{1, \ldots, m\}$

$$A_j = \sum_{i:y_{ij}=1} x_i x_i^\top, \quad b_j = \sum_{i:y_{ij}=1} x_i, \quad G = \sum_{i=1}^{n} x_i x_i^\top \tag{6}$$

The exact solution is then given by

$$\theta_j = [A_j + \alpha G + \lambda I]^{-1} b_j.$$

Notice that the solution $\theta_j$ for class $j$ depends on class-specific statistics (the matrix $A_j$ and the vector $b_j$), as well as the global quantity $G$. Algorithm 2 computes a private estimate of the solution by adding Gaussian noise to each of $G, A_j, b_j$ (Lines 2, 4, and 5 respectively), then solving the linear system using the noised statistics (Line 6). This is a variant of the popular sufficient statistics perturbation algorithm for DP linear regression. One crucial observation is that the noisy version of $G$ is only computed once (Line 2), and reused for all classes. This allows to control the sensitivity of the solution w.r.t. each example; indeed, suppose example $i$ only has one positive class $j_0$, then that example only contributes to $G$, $A_{j_0}$, and $b_{j_0}$. In particular, the sensitivity of the solution (and hence the amount of noise we need to add) does not scale with the total number of classes, only with the number of *positive* classes per example. This is represented by the parameter $k$ in Algorithm 2 ($k$ is equal to 1 for single-class classification tasks, and even in multi-class tasks, we typically have $k \ll m$). We now give the formal privacy guarantee:

**Theorem 2.2** (Privacy guarantee for Algorithm 2)**.** *Algorithm 2 satisfies $\frac{3}{2\sigma^2}$-zCDP.*

*Proof.* First, let $D, D'$ be neighboring data sets that differ in the data point $(x, y)$, and let $G(D)$ be the global statistic in equation (6) computed on data set $D$. Then $\|G(D) - G(D')\|_F = \|\tilde{x}\tilde{x}^\top\|_2$, and

since $\|\tilde{x}\|_2 \leq C$ (due to clipping), $\|G(D) - G(D')\|_F \leq C^2$, thus the computation of $G$ (Line 2) is $\frac{1}{2\sigma^2}$-zCDP. Now moving to the computation of $A_j$: let $A(D)$ be the concatenation of all the class-specific statistics, i.e. $A(D) = [A_1(D)|\ldots|A_m(D)]$. Notice that $\|A_j(D) - A_j(D')\|_F = \|\tilde{x}\tilde{x}^\top\|_2$ if $y_j = 1$ and 0 otherwise. Since by assumption, the number of positive classes per example is bounded by $k$, we have that $\|A(D) - A(D')\|_F \leq \sqrt{k}\|\tilde{x}\tilde{x}^\top\|_F$, which is bounded by $\sqrt{k}C^2$ due to clipping. Therefore the computation of all $A_j$ combined (Line 4) is $\frac{1}{2\sigma^2}$-zCDP. Finally, by a similar argument, we have that $\|b(D) - b(D')\|_2 \leq \sqrt{k}C$, and the computation of all $b_j$ combined (Line 5) is $\frac{1}{2\sigma^2}$-zCDP.

By simple composition of zCDP Bun & Steinke (2016), the algorithm is $\frac{3}{2\sigma^2}$-zCDP. $\qquad\square$

---

**Algorithm 2** Differentially Private Least Squares

---

**Require:** Data set $D = \{(x_1, y_1), \cdots, (x_n, y_n)\}$ with $(x_i, y_i) \in \mathcal{D}$, weight coefficient $\alpha$, regularization coefficient $\lambda$, maximum number of positive classes per example: $k$, clipping norm: $C$, noise multiplier: $\sigma$.

1: Clip all features: $\tilde{x}_i \leftarrow \mathsf{clip}(x_i)$ for all $i \in \{1, \ldots, n\}$.

2: $\tilde{G} \leftarrow \sum_{i=1}^{n} \tilde{x}_i\tilde{x}_i^\top + \mathcal{N}\left(\vec{0}_{d\times d}, \left(\sigma C^2\right)^2\right)$, where $\mathcal{N}\left(\vec{0}_{d\times d}, (\sigma C^2)^2\right)$ indicates a $d \times d$-matrix, each of whose coordinates is an i.i.d. Gaussian with standard deviation $\sigma C^2$.

3: **for** $j = 1, \ldots, m$ **do**

4: $\quad \tilde{A}_j \leftarrow \sum_{i:y_{ij}=1} \tilde{x}_i\tilde{x}_i^\top + \mathcal{N}\left(\vec{0}_{d\times d}, \left(\sigma\sqrt{k}C^2\right)^2\right)$

5: $\quad \tilde{b}_j \leftarrow \sum_{i:y_{ij}=1} \tilde{x}_i + \mathcal{N}\left(\vec{0}_d, (\sigma\sqrt{k}C)^2\right)$.

6: $\quad \theta_j \leftarrow \left[\tilde{A}_j + \alpha\tilde{G} + \lambda I\right]^{-1}\tilde{b}_j$

7: **end for**

8: **return** $\theta$.

---

## 2.4  DP-FC

From our early experiments, we found that Newton's method with logistic regression performs better than least squares linear regression in non-private setting. But in the private setting, DP-LS outperforms DP-Newton, especially for lower values of epsilons. Notice that both are second-order methods, and both rely on estimating and inverting the Hessian (Line 9 in Algorithm 1 and Line 6 in Algorithm 2); the main advantage of DP-LS is that the private hessian computation does not have to composed over classes or iterations. In this section, in order to mitigate the trade-off between the two methods, we introduce a method called Differentially Private SGD with Feature Covariance (DP-FC) which leverages covariance of features to make use of second-order information without paying the cost of composition over classes or iterations. Indeed, since feature covariance neither depends on the model parameters nor the prediction, it can be shared across both classes and iterations. This is described in Algorithm 3. The method can be interpreted as DP-SGD with preconditioning (where the approximate feature covariance $\tilde{G}$ is used as preconditioner). We empirically observe that this leads to greatly reduced sensitivity compared to DP-Newton and significant improvements in overall metrics across all values of epsilons we tried.

**Theorem 2.3** (Privacy guarantee for Algorithm 3). *Algorithm 3 satisfies $\frac{(T+1)}{2\sigma^2}$-zCDP.*

*Proof.* The crux of the proof is to ensure that the computation of $\tilde{G}$ (Line 3) and $\tilde{g}_t$ (Line 7) individually satisfy $\frac{1}{2\sigma^2}$-zCDP. The rest of the proof is by simple composition of zCDP Bun & Steinke (2016) (once for the $\tilde{G}$ computation, and $T$ times for the $\tilde{g}_t$ computation).

Let $D$ and $D'$ be neighboring datasets and let $(x, y)$ be the differing datapoint between $D$ and $D'$.

In the computation of $g_t$, we have the following: $|g_t(D) - g_t(D')| = |\mathsf{clip}(\nabla\ell(\theta_t; (x, y))| \leq C_g$. It immediately follows that the computation of $\tilde{g}_t$ for each $t \in \{0, \ldots, T-1\}$ satisfies $\frac{1}{2\sigma^2}$-zCDP. Now, moving on to the sensitivity of $G$. We have $\|G(D) - G(D')\| = \|\tilde{x}\tilde{x}^\top\| \leq C_G^2$, since $\|\tilde{x}\|_2 \leq C_G$. It immediately follows that $\tilde{G}$ satisfies $\frac{1}{2\sigma^2}$-zCDP, which completes the proof. $\qquad\square$

---

**Algorithm 3** Differentially Private SGD with Feature Covariance (DP-FC) Method

---

**Require:** Data set $D = \{(x_1, y_1), \cdots, (x_n, y_n)\}$ with $(x_i, y_i) \in \mathcal{D}$, loss function: $\ell : \mathbb{R}^{m \times d} \times \mathbb{R} \to \mathbb{R}$, learning rate $\eta$, clipping norms: $C_G$ and $C_g$, number of iterations: $T$, noise multiplier: $\sigma$

1: Clip features for covariance computation: $\tilde{x}_i \leftarrow \mathsf{clip}(x_i)$ for all $i \in \{1, \ldots, n\}$ where $\mathsf{clip}(x) = x \min\left\{1, \frac{C_G}{\|x\|_2}\right\}$.

2: Compute the feature covariance: $G = \sum_{i=1}^{n}(\tilde{x}_i \tilde{x}_i^\top)$.

3: $\tilde{G} \leftarrow \frac{1}{n} G + \mathcal{N}\left(\vec{0}_{d \times d}, (\frac{\sigma C_G^2}{n})^2 I_d\right) + \lambda I$

4: Randomly initialize $\theta_0$.

5: **for** $t = 1, \ldots, T$ **do**

6:  $g_t \leftarrow \sum_{i=i}^{n} \mathsf{clip}(\nabla \ell(\theta_t; (x_i, y_i)))$ where $\mathsf{clip}(g) = g \cdot \min\left\{1, \frac{C_g}{\|g\|_2}\right\}$.

7:  $\tilde{g}_t \leftarrow \frac{1}{n} g_t + \mathcal{N}\left(\vec{0}_d, (\frac{\sigma C_g}{n})^2\right)$.

8:  $\theta_{t+1} \leftarrow \theta_t - \eta \tilde{g}_t \tilde{G}^{-1}$

9: **end for**

10: **return** $\theta_T$.

---

## 3 Empirical Results

| Pretraining | Method | Epochs | NP | Epsilon | | | | | | | |
| --- | --- | --- | --- | --- | --- | --- | --- | --- | --- | --- | --- |
| | | | | 0.01 | 0.05 | 0.1 | 0.5 | 1.0 | 2.0 | 4.0 | 8.0 |
| ImageNet-21K | DP-LS | 1 | 75.5 | 71.2 | 71.5 | 71.5 | 73.0 | 73.6 | 74.1 | 74.6 | 75.0 |
| | DP-Newton | 1 | 72.7 | 70.2 | 71.3 | 71.3 | 71.4 | 71.6 | 71.7 | 72.0 | 72.2 |
| | | 10 | 78.2 | 66.5 | 71.0 | 71.4 | 71.9 | 71.9 | 72.3 | 73.0 | 74.2 |
| | DP-FC | 1 | 72.9 | **71.9** | **72.1** | **72.5** | 72.9 | 72.9 | 72.9 | 72.9 | 72.9 |
| | | 10 | 77.1 | 71.8 | 71.9 | 72.1 | **75.4** | **76.3** | **76.8** | **77.0** | **77.1** |
| | DP-Adam | 1 | 73.8 | - | - | - | - | 39.1 | 54.2 | 63.3 | 68.3 |
| | | 10 | 76.1 | - | - | 22.1 | 52.9 | 62.3 | 66.6 | 69.5 | 71.0 |
| | | 100 | **78.3** | - | - | - | 47.2 | 59.1 | 65.8 | 68.8 | 70.4 |
| JFT | DP-LS | 1 | 87.5 | **82.4** | **83.8** | 84.1 | 85.8 | 86.2 | 86.4 | 86.6 | 86.7 |
| | DP-Newton | 1 | 85.9 | 77.8 | 80.3 | 81.0 | 82.9 | 83.6 | 84.0 | 84.5 | 84.9 |
| | | 10 | **88.9** | 76.0 | 79.7 | 80.1 | 81.8 | 82.9 | 83.1 | 84.7 | 85.3 |
| | DP-FC | 1 | 85.8 | 82.1 | 83.8 | **84.3** | 85.1 | 85.4 | 85.5 | 85.6 | 85.6 |
| | | 10 | 88.4 | 81.0 | 83.1 | 83.7 | **86.1** | **86.8** | **87.4** | **87.8** | **88.0** |
| | DP-Adam | 1 | 84.8 | - | 68.7 | 74.0 | 82.1 | 83.7 | 84.4 | 84.8 | 84.8 |
| | | 10 | 87.3 | - | 70.5 | 75.6 | 83.4 | 84.9 | 85.6 | 86.3 | 86.7 |
| | | 100 | 88.6 | - | 49.7 | 64.4 | 78.3 | 81.5 | 83.9 | 85.4 | 86.3 |

Table 2: Comparison of Top-1 test accuracies when privately fine-tuning on Imagenet-1K. We denote accuracy $\leq 20\%$ with the symbol '-'. When pre-trained with JFT, we observe that DP-FC performs best for epsilon values ranging from $[0.1, 8.0]$ whereas DP-LS is best for even lower epsilons. In the case of pre-training with ImageNet-21k, we find that DP-FC (10 epochs) outperforms all other methods across the board. For reference, we note that best non-private accuracy on ImageNet-1k is 91% (Yu et al., 2022b). We used ViT-B/16 model for pre-training with ImageNet-21k and ViT-G/14 for JFT.

**Datasets.** We use 3 datasets for private finetuning, namely 1) ImageNet-1k (Deng et al., 2009) with 1k classes and 1.3M images 2) CIFAR-10 and 3) CIFAR-100. We also refer to these as the private dataset for which we want a privacy guarantee. For pre-training, we rely on JFT-3B, ImageNet-21k and ImageNet-1K

| Pretraining | Method | Epochs | Non-Private | Epsilon | | | | | | | |
|---|---|---|---|---|---|---|---|---|---|---|---|
| | | | | 0.01 | 0.05 | 0.1 | 0.5 | 1.0 | 2.0 | 4.0 | 8.0 |
| ImageNet-1K | DP-LS | 1 | 91.3 | 81.1 | 83.7 | 84.3 | 86.3 | 87.7 | 88.7 | 89.6 | 90.3 |
| | DP-Newton | 1 | 91.0 | 77.4 | 79.6 | 80.5 | 84.2 | 85.7 | 87.1 | 88.4 | 89.2 |
| | | 10 | **91.6** | 79.7 | 81.4 | 82.7 | 86.1 | 87.5 | 89.0 | 88.6 | 90.2 |
| | DP-FC | 1 | 91.0 | 79.0 | 81.5 | 83.1 | 86.6 | 88.0 | 88.9 | 89.7 | 90.3 |
| | | 10 | 91.1 | **81.2** | **84.7** | **86.5** | **89.5** | **90.4** | **91.0** | **91.1** | **91.1** |
| | DP-Adam | 1 | 77.9 | 52.6 | 74.3 | 76.7 | 78.2 | 78.2 | 77.8 | 77.6 | 77.8 |
| | | 10 | 82.3 | 56.4 | 78.3 | 76.7 | 82.0 | 82.6 | 82.4 | 82.2 | 82.7 |
| | | 100 | 87.5 | 40.9 | 63.8 | 72.3 | 83.6 | 86.8 | 87.4 | 87.4 | 87.6 |
| ImageNet-21K | DP-LS | 1 | 96.5 | 94.5 | 95.2 | 95.4 | 95.8 | 96.0 | 95.5 | 96.2 | 96.3 |
| | DP-Newton | 1 | 96.5 | 94.3 | 94.9 | 95.2 | 95.6 | 95.7 | 96.0 | 96.1 | 96.2 |
| | | 10 | 96.5 | 94.8 | 95.2 | 95.4 | 95.6 | 95.9 | 96.0 | 96.2 | 96.2 |
| | DP-FC | 1 | 96.6 | 94.6 | 95.2 | 95.5 | 95.9 | 96.0 | 96.2 | 96.3 | 96.3 |
| | | 10 | **96.6** | **94.8** | **95.6** | **95.8** | **96.1** | **96.3** | **96.5** | **96.5** | **96.5** |
| | DP-Adam | 1 | 95.2 | 90.0 | 94.7 | 95.1 | 95.1 | 95.1 | 95.1 | 95.1 | 95.2 |
| | | 10 | 96.1 | 83.8 | 94.9 | 95.5 | 95.8 | 95.8 | 95.8 | 95.9 | 96.0 |
| | | 100 | 96.5 | 64.0 | 90.0 | 93.3 | 95.5 | 95.7 | 95.9 | 96.1 | 96.2 |
| JFT | DP-LS | 1 | 98.9 | **97.4** | 98.2 | 98.4 | 98.4 | 98.6 | 98.8 | 98.8 | 98.9 |
| | DP-Newton | 1 | 98.9 | 94.1 | 96.5 | 97.2 | 98.1 | 98.3 | 98.5 | 98.7 | 98.8 |
| | | 10 | 98.9 | 95.9 | 97.5 | 97.9 | 98.2 | 98.5 | 98.6 | 98.4 | 98.8 |
| | DP-FC | 1 | **98.9** | 95.2 | 97.6 | 97.9 | 98.5 | 98.6 | 98.8 | 98.8 | **98.9** |
| | | 10 | 98.9 | 97.3 | **98.2** | **98.4** | **98.8** | **98.8** | **98.9** | **98.9** | 98.9 |
| | DP-Adam | 1 | 97.5 | 93.5 | 97.0 | 97.5 | 97.5 | 97.6 | 97.6 | 97.6 | 97.6 |
| | | 10 | 98.7 | 87.6 | 97.7 | 98.1 | 98.5 | 98.6 | 98.7 | 98.7 | 98.7 |
| | | 100 | 98.9 | 79.2 | 93.2 | 96.3 | 98.3 | 98.6 | 98.6 | 98.8 | 98.8 |

Table 3: Comparison of Top-1 test accuracies when private finetuning on CIFAR-10. We denote accuracy $\leq$ 20% with the symbol '-'. Similar to other datasets, DP-FC (10 epochs) outperform all other methods almost across the board with a single exception of epsilon of 0.01 when pre-training with JFT were DP-LS performs slightly better. For reference, we note that best non-private accuracy on CIFAR-10 is 99.5% (Dosovitskiy et al., 2021). We used ViT-B/16 model for pre-training with ImageNet-21k and Imagenet-1K, and ViT-G/14 for JFT.

(as done in Zhai et al. (2021)). For JFT, we intentionally chose a slightly smaller version of the dataset i.e. JFT-3B instead of JFT-4B, enabling us to exactly follow Zhai et al. (2021) and thus lowering the risk of the project. Also note that, as done in recent works, none of our finetuning datasets in reality are sensitive datasets: we are only simulating a public/private dataset split only for demonstration purposes (Kurakin et al., 2022; Mehta et al., 2022; De et al., 2022). The JFT datasets are not publicly available but have been used extensively as a pre-training dataset in the non-private setting to obtain state-of-the-art results (Dosovitskiy et al., 2021; Brock et al., 2021; Tolstikhin et al., 2021; Zhai et al., 2021). Similar to Mehta et al. (2022); De et al. (2022), to make sure that our simulated "public" and "private" datasets capture a practical scenario, we carefully de-duplicate our pre-training datasets w.r.t. **all** splits of our finetuning datasets (Kolesnikov et al., 2020; Dosovitskiy et al., 2021). More details about this process can be found in the appendix.

**Model variants.** We evaluate the transfer learning capabilities of the Vision Transformer (ViT) (Dosovitskiy et al., 2021) model family in our study. We follow the standard notation to indicate the model size and the

| Pretraining | Method | Epochs | Non-Private | Epsilon | | | | | | | |
|---|---|---|---|---|---|---|---|---|---|---|---|
| | | | | 0.01 | 0.05 | 0.1 | 0.5 | 1.0 | 2.0 | 4.0 | 8.0 |
| ImageNet-1K | DP-LS | 1 | 71.9 | **49.2** | **51.8** | **53.9** | 57.6 | 60.0 | 62.5 | 65.4 | 67.5 |
| | DP-Newton | 1 | 68.8 | 44.6 | 48.6 | 49.3 | 50.2 | 51.2 | 54.4 | 57.2 | 59.5 |
| | | 10 | **72.1** | 36.4 | 48.2 | 49.8 | 50.7 | 51.8 | 53.5 | 56.1 | 59.5 |
| | DP-FC | 1 | 68.7 | 49.2 | 50.2 | 52.1 | 58.3 | 60.8 | 62.8 | 64.6 | 66.1 |
| | | 10 | 71.4 | 48.9 | 49.7 | 53.7 | **61.4** | **64.9** | **68.2** | **69.8** | **70.4** |
| | DP-Adam | 1 | 52.1 | - | 26.0 | 34.6 | 47.6 | 51.2 | 51.7 | 51.7 | 52.2 |
| | | 10 | 57.3 | - | 20.0 | 33.5 | 51.2 | 54.4 | 55.6 | 57.5 | 56.3 |
| | | 100 | 67.9 | - | - | - | 36.3 | 40.2 | 51.5 | 58.4 | 63.7 |
| ImageNet-21K | DP-LS | 1 | 83.9 | **77.2** | **77.7** | 78.1 | 79.8 | 80.7 | 81.4 | 81.9 | 82.4 |
| | DP-Newton | 1 | 83.0 | 76.5 | 77.2 | 77.2 | 77.6 | 78.3 | 79.0 | 79.5 | 80.5 |
| | | 10 | 83.0 | 73.6 | 77.4 | 77.8 | 78.3 | 78.9 | 79.6 | 80.4 | 81.4 |
| | DP-FC | 1 | 83.0 | 77.1 | 77.5 | 78.2 | 80.0 | 80.9 | 81.6 | 81.9 | 82.4 |
| | | 10 | 84.3 | 77.1 | 77.1 | **78.5** | **81.6** | **82.7** | **83.3** | **83.8** | **83.9** |
| | DP-Adam | 1 | 79.9 | 27.6 | 67.0 | 72.7 | 78.3 | 79.5 | 79.7 | 79.7 | 79.7 |
| | | 10 | 82.0 | - | 55.7 | 67.5 | 78.6 | 80.7 | 81.5 | 81.5 | 81.6 |
| | | 100 | **84.6** | - | 29.7 | 45.2 | 71.3 | 76.2 | 79.5 | 81.1 | 82.2 |
| JFT | DP-LS | 1 | **90.6** | **74.9** | **80.3** | **82.5** | 85.5 | 86.4 | 87.7 | 88.4 | 88.9 |
| | DP-Newton | 1 | 89.9 | 73.1 | 72.6 | 73.4 | 78.5 | 80.9 | 82.8 | 84.6 | 85.9 |
| | | 10 | 89.9 | 69.6 | 75.7 | 76.7 | 77.4 | 77.6 | 80.0 | 82.9 | 85.4 |
| | DP-FC | 1 | 89.9 | 73.5 | 78.6 | 81.0 | 85.2 | 86.7 | 87.7 | 88.3 | 88.6 |
| | | 10 | 90.1 | 72.1 | 75.9 | 79.0 | **86.2** | **88.1** | **89.0** | **90.0** | **90.1** |
| | DP-Adam | 1 | 83.5 | 27.9 | 61.8 | 71.3 | 79.7 | 82.1 | 83.4 | 83.5 | 83.5 |
| | | 10 | 88.2 | 21.9 | 60.2 | 69.7 | 81.9 | 83.9 | 86.2 | 86.8 | 87.8 |
| | | 100 | 90.0 | - | 29.4 | 50.5 | 73.7 | 78.7 | 83.1 | 86.1 | 88.0 |

Table 4: Comparison of Top-1 test accuracies when private finetuning on CIFAR-100. We denote accuracy ≤ 20% with the symbol '-'. Similar to other datasets, DP-FC outperforms all other methods for moderate privacy budgets whereas DP-LS performs slightly better for very strict privacy guarantees depending on the pre-training dataset. For reference, we note that best non-private accuracy on CIFAR-100 is 96.08% (Foret et al., 2021). We used ViT-B/16 model for pre-training with ImageNet-21k and Imagenet-1K, and ViT-G/14 for JFT.

input patch size, for example, ViT-B/32 means the "Base" variant with 32x32 input patch size. Note that for ViT, compute requirements scales up as we reduce the patch size. We obtained features from ViT-G/14 model pre-trained on JFT-3B (Zhai et al., 2021), and ViT-B/16 pre-trained on ImageNet-21k and ImageNet-1k. (Steiner et al., 2021).

Next, we present our main set of private fine-tuning results and core observations on all 3 datasets, namely ImageNet-1k (Table 2), CIFAR-10 (Table 3) and CIFAR-100 (Table 4).

### 3.1 Better pre-training continues to improve private fine-tuning performance

We evaluate private fine-tuning performance on features extracted from pre-trained models of 2 sizes i.e. ViT-G/14 and ViT-B/16, pre-trained on 3 different datasets, namely JFT-3B, ImageNet-21K and ImageNet-1K. We do this to quantify the extent to which the representation quality is improved by increasing pre-training dataset in combination of model size. As shown in Figure 1, as the model size and pre-training dataset size is

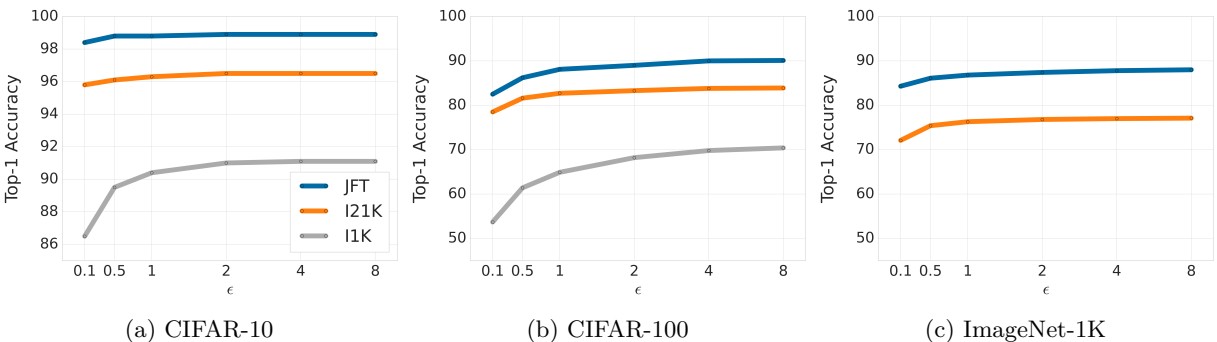

Figure 1: Comparison of top-1 accuracies with private fine-tuning using DP-FC method on all 3 datasets across a range of epsilons. We observe that better pre-training helps even more for lower values of epsilon (stricter privacy guarantee).

increased, we continue to see improvement in downstream private fine-tuning performance for all 3 datasets we consider. In addition, we make following observations from our results.

First, **better pre-training can help even more at stricter privacy budgets**. As shown in Figure 1, for both CIFAR-10 and CIFAR-100, when comparing features extracted from ViT-B/16 pre-trained with ImageNet-1K and features from ViT-G/14 pre-trained with JFT, the improvement in performance at $\varepsilon = 1$ is larger than $\varepsilon = 8$. We see a similar trend at even lower epsilons in Table 3 and Table 4.

Second, **features extracted from off-the-shelf pre-trained models can suffice for DP**. Except for the fact that we deduplicate all splits of our pre-training dataset with our fine-tuning datasets, we use the exact same procedure used to pre-train large vision models. This suggests that, in practice, a recipe where features extracted from large off-the-shelf vision model used to privately fine-tune a classifier can be quite effective for DP performance. Since there is no need for a special pre-trained model for use in DP, this considerably reduces the cost of training private image classifiers.

Lastly, **private fine-tuning of high quality features closes the gap between private and non-private performance considerably**. In the non-private setting, Zhai et al. (2021) obtain an impressive 90.45% top-1 accuracy by fine-tuning the whole ViT-G/14 model on ImageNet-1K dataset. We observe that even by fine-tuning just the last layer, we can obtain as much as 88.9% top-1 accuracy on ImageNet-1K from the same sized model. Thus the marginal benefit of fine-tuning the whole model is <2% even in the non-private case. In the private case, fine-tuning of pre-extracted features with DP-FC at $\varepsilon = 8$ leads to state of the art 88% top-1 accuracy. On ViT-G/14, this represents $< 2.5\%$ difference between best non-private and private performance. This is also just 3% below the best non-private accuracy of 91% on ImageNet-1K (Yu et al., 2022b).

### 3.2 Better optimizers improve privacy-utility trade-off

We observe that the choice of optimizer can have a significant impact on the privacy-utility trade-off.

First, **optimizers that work well in the large batch regime are better for private training**. Even in the non-private setting, it is known that batch size is intimately tied to the optimization procedure and can lead to suboptimal use of resources if increased beyond a certain point while keeping number of epochs constant (Goyal et al., 2017; You et al., 2017; 2019). The maximum batch size which can be used without jeopardizing the utility-compute trade-off is also heavily dependent on the choice of optimizer (Zhang et al., 2019). Though in the private setting, several works have observed that increasing the batch size leads to improved privacy-utility trade-off (Dormann et al., 2021; Li et al., 2014; Hoory et al., 2021). It is also empirically observed that the utility in large batch regime can be further improved by leveraging optimizers which work well in large batch regime, such as DP-Adam or DP-LAMB (Anil et al., 2021; Mehta et al., 2022; Bu et al., 2022a). Second-order methods such as LS or Newton are also well-suited to the large-batch regime (they were developed and are typically used in the full-batch setting).

Second, **optimizers with faster convergence rates are advantageous in the private setting**, because they can reduce the number of epochs required for convergence. Indeed, in typical privacy analysis of iterative methods, the analysis works by composition over iterations, which means that the privacy penalty scales with the number of visitations of the data. Thus, in addition to the computational benefit, any improvement in convergence rate directly helps training with privacy because it requires less noise to be added under the same privacy constraints. We empirically observe that this reduction in noise in the case of DP-FC, in combination with sharing of the privatized Feature Covariance across classes and iterates, allow us to obtain better results with 10 epochs compared to even when using 100 epochs with DP-SGD.

## 4    Related Work

Differential privacy (Dwork et al., 2006b) is a popular method to guarantee privacy in a quantifiable way in many data-driven applications. To achieve differential privacy in machine learning, tasks practitioners commonly train models with privatized variation of gradient descent, called DP-SGD (Song et al., 2013; Bassily et al., 2014; Abadi et al., 2016).

Despite theoretical guarantees, differentially private training has two major drawbacks which limits its wide adoption. First of all, DP-SGD can be much slower compared to regular SGD, if implemented naively. To address this, several deep learning architectures offer to vectorize the computation (Subramani et al., 2020) or it is sometimes possible to bound the sensitivity of each example without calculating the gradient for every example separately, leading to a dramatic cost-reduction both in terms of memory and compute (Goodfellow, 2015; Li et al., 2022b; Bu et al., 2022a;d). Although, in our work this cost is minimal since we only train the last layer. We do note that computational burden can still be an issue for optimization schemes like DP-Newton where we had to resort to feature clipping in order to produce bounds on sensitivity of the gradient and the hessian.

In addition to the computational cost, model trained with differentially privacy usually suffer from so-called "utility loss", which means that accuracy (or any other quality metric) is worse (and sometimes significantly worse) compared to accuracy of non-private model (Dormann et al., 2021; Klause et al., 2022). Over the years, several lines of improvements have been proposed including adaptive clipping (Pichapati et al., 2019; Thakkar et al., 2019; Bu et al., 2022b; Golatkar et al., 2022), param-efficient finetuning (Yu et al., 2022a; Mehta et al., 2022; Bu et al., 2022c; Cattan et al., 2022; Li et al., 2022a) and even leveraging intermediate checkpoints (De et al., 2022; Shejwalkar et al., 2022). One of the recent trends to improve utility of private models significantly involves various ideas related to transfer learning where previous works demonstrate improved performance in the setting where we have access to a large public or non-sensitive dataset of the same modality as the private data (Kurakin et al., 2022; De et al., 2022; Mehta et al., 2022; Tramèr & Boneh, 2021; Yu et al., 2022a; Li et al., 2022b; Kurakin et al., 2022; Hoory et al., 2021). Our work also leverages large pre-trained models in order to obtain high-quality features for private finetuning. In addition, similar to the works that focus on studying and reducing dimensionality of the model in the context of DP (Li et al., 2022a; Golatkar et al., 2022; Yu et al., 2021; Zhang et al., 2021; Zhou et al., 2020), we focus solely on learning just the last layer privately, which can be seen as an implicit way to reduce dimensionality.

In the context of differential privacy, several recent papers also advocate the use of large batch sizes in order to improve the privacy-utility tradeoff (Mehta et al., 2022; McMahan et al., 2018; Anil et al., 2021; Dormann et al., 2021; Hoory et al., 2021; Liu et al., 2021b; Kurakin et al., 2022). Even though this work explicitly does not explore the affect changing the batch size, the fact that we are able to obtain state of the art results in the full batch setting may point to the effectiveness of large batch sizes in the context of DP. Further, our work also zooms in on differentially private linear and logistic regression. Several existing works have studied differential private convex optimization (Chaudhuri et al., 2011; Kifer et al., 2012; Song et al., 2013; Bassily et al., 2014; Wu et al., 2016; McMahan et al., 2017; Bassily et al., 2019; Iyengar et al., 2019; Feldman et al., 2020; Bassily et al., 2020; Song et al., 2020; Andrew et al., 2021). There is also a growing interest in the special case of linear regression (Smith et al., 2017; Sheffet, 2019; Liu et al., 2021a; Cai et al., 2021; Varshney et al., 2022; Wang, 2018) and even second order methods in the context of differential privacy (Avella-Medina et al., 2021; Chien et al., 2021). In this work, we illustrate empirically the extent to which second order

methods can help in DP. It would be quite interesting to see how other popular second order methods like Shampoo (Gupta et al., 2018; Anil et al., 2020) would fare in the context of DP.

## 5 Conclusion

In this work, we focus on private finetuning of image classification datasets using features extracted from a pre-trained model. Given that finetuning just the features is significantly cheaper than finetuning the full model, we systematically explore optimization schemes which are perceived to be expensive in high-dimensional settings. As illustrated on 3 finetuning datasets i.e. ImageNet-1k, CIFAR-10 and CIFAR-100, we find that DP-LS (Least Squares) outperforms DP-SGD with logistic regression, especially for lower values of epsilons. Given the intuition that 2nd order information may be the reason for superior performance of DP-LS, we also explore Newton's method with Logistic Loss. Noticing that the amount of noise required by Newton's method scales with the number of classes and iterations, we introduce an optimization scheme called DP-FC which replaces the hessian by the feature covariance matrix, that can be shared across classes and iterations. Using this insight, we demonstrate that it is indeed possible to get state of the art results by just finetuning the last layer of a pre-trained model with privacy constraints. We hope that our work significantly reduces the barrier in training private models.

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

## A  Limitations

This work leverages a large properietary dataset called JFT-3B to pre-train ViT-G/14 model in order to illustrate the benefits of scale on differential privacy with transfer learning. In order to make our work more generalizable and reproducible, we also include results with models pre-trained with ImageNet-21k and ImageNet-1k.

Another limitation of our work may be the fact that our pre-training dataset is largely in-distribution with the private fine-tuning datasets. We would like to argue that, in practice, this is still valuable since it illustrates the effectiveness of the approach, and helps estimate the utility of gathering a public dataset to pre-train on, given a sensitive dataset that one wants privacy guarantee over. Finally, out of distribution performance is an interesting research question even in the non-private setting and its exploration in the context of privacy can be a direction of very valuable future work.

We additionally note that we do not include any cost related to hyperparmeter tuning in our privacy accounting and budget. This is in line with the setup of the baselines we consider and allows for a fair comparison with reported numbers. However, we recognize this as a limitation of our work and more research in this direction a considerable opportunity.

In terms of societal impact, the biggest cost of this work is training the largest ViT-G model on a large dataset and its energy impact. However, we argue that our results ultimately point towards amortizing and increasingly leveraging already trained models for high-performance DP training, and thus potentially reducing the overall energy consumption.

## B  Algorithmic details

### B.1  Privacy Analysis details for DP-SGD

---

**Algorithm 4** Generalized First Order Differentially Private Algorithm

---

**Require:** Data set $D = \{(x_1, y_1), \cdots, (x_n, y_n)\}$ with $(x_i, y_i) \in \mathcal{D}$, loss function: $\ell : \mathbb{R}^{m \times d} \times \mathbb{R} \to \mathbb{R}$, a first order optimizer $Opt$, clipping norm: $C$, number of iterations: $T$, noise multiplier: $\sigma$

1: Randomly initialize $\theta_0$.
2: **for** $t = 1, \ldots, T$ **do**
3:    $g_t \leftarrow \frac{1}{n} \sum_{i=1}^{n} \mathsf{clip}\left(\nabla \ell(\theta_t; (x_i, y_i))\right)$, where $\mathsf{clip}(v) = v \cdot \min\left\{1, \frac{C}{\|v\|_2}\right\}$.
4:    $\tilde{g}_t \leftarrow g_t + \mathcal{N}\left(0, (\sigma C)^2\right)$
5:    $\theta_{t+1} \leftarrow$ single step of first order optimization with gradient $Opt(\tilde{g}_t)$
6: **end for**
7: **return** $\frac{1}{T} \sum_{t=1}^{T} \theta_t$ or $\theta_T$.

---

The privacy parameters $(\varepsilon, \delta)$ are functions of $C$, $\sigma$, $|B_t|$, $|\mathcal{D}|$, and the total number of iterations $T$. DP-SGD algorithm involves setting the right clipping norm $C$ and the noise multiplier $\sigma$ given a privacy budget, batch and dataset size. The $(\varepsilon, \delta)$ guarantee is computed by analysis of the Gaussian Mechanism with privacy amplification by subsampling and composition across across iterations (Kasiviswanathan et al., 2008; Bassily et al., 2014; Abadi et al., 2016; Mironov, 2017; McMahan et al., 2017; Mironov et al., 2019; Erlingsson et al., 2019; Zhu & Wang, 2019; Feldman et al., 2020; Wang et al., 2020). Our implementation relies on Tensorflow Privacy [1] codebase for conversion of $(\varepsilon, \delta)$ and clipping norm $C$ to/from noise multiplier $\sigma$. We rely on the default Rényi accountant implementation already open-sourced as part of Tensorflow Privacy library.

To put the epsilon-delta values in context, privacy guarantee for let's say $\varepsilon \approx 4$ on ImageNet-1K satisfies a much stronger property of zCDP $\leq 1$ (0.154 for $\varepsilon = 4$) which is by now an industry standard.

## C  Pre-training Details

We conduct all our experiments in Jax (Bradbury et al., 2018; Frostig et al., 2018) is framework that leverages just-in-time compilation using XLA[2] and does auto-vectorization of the backward pass. We leverage this functionality throughout our experiments. Finally, we conduct our experiments on TPUv4 architecture.

### C.1  Pre-training with JFT-3B

**Dataset.** Contrary to Mehta et al. (2022); De et al. (2022), we use a smaller version of JFT, namely JFT-3B (instead of JFT-4B) for our pre-training. We do this to lower the risk of the project and follow Zhai et al. (2021) exactly for pre-training ViT-G/14 model. JFT-3B dataset consists of nearly 3 billion images, annotated with a class-hierarchy of around 30k labels via a semiautomatic pipeline. As done previously, we ignore the

---

[1]https://github.com/tensorflow/privacy
[2]https://www.tensorflow.org/xla

hierarchical aspect of the labels and use only the assigned labels as targets for multi-label classification via a sigmoid cross-entropy loss.

**Deduplication.** In order to both not inflate our results and break privacy guarantee offered by fine-tuning privately on ImageNet, we extend the deduplication process proposed by Kolesnikov et al. (2020) and deduplicate both JFT-3B with respect to all splits of ImageNet. We use a model based deduplication system which removes both exact and near-duplicates across common image transformation like crop, shift, resize etc.

**Hyperparameters.** At the pre-training stage, we follow Zhai et al. (2021) exactly and stick with the common practice of employing Adafactor optimizer with $\beta_1 = 0.9$ and $\beta_2 = 0.999$, with a batch size of 32768, dropout rate of 0.0, clip global norm of 1, and a high weight decay of 3.0 for the "head" and 0.03 for the "body". In addition, we remove the additional [$class$] token to save memory. Finally, all the models are pre-trained at resolution [224, 224], with inception crop followed by random horizontal flip pre-process. We also use reciprocal square-root schedule with a linear learning rate warmup of 10k steps. Finally, ViT-G/14 model was pre-trained using 2048 TPUv3 chips.

## C.2 Pre-training with ImageNet21k and ImageNet1k

**Datasets.** ImageNet-21k is a superset of ImageNet-k with 21k classes and 14M images (Deng et al., 2009). Similar to before, in order to both not inflate our results and break privacy guarantee, we extend the deduplication process proposed by Kolesnikov et al. (2020) and deduplicate ImageNet-21k with respect to all splits of ImageNet-1k, CIFAR-10 and CIFAR-100. Similarly, we deduplicate ImageNet-1k with respect to all splits of CIFAR-10 and CIFAR-100.

**Hyperparameters.** At the pre-training stage, we stick with the common practice of employing Adam optimizer (even for ResNet) with $\beta_1 = 0.9$ and $\beta_2 = 0.999$, with a batch size of 4096. Unlike pre-training with JFT dataset, we follow recommendations from Steiner et al. (2021) to use AugReg strategy where we lower the weight decay to 0.1 (which gets multiplied by the learning rate) and don't use dropout but instead use data augmentation strategy called **medium1** which combines Mixup with $\alpha = 0.2$ (Zhang et al., 2017) and RandAugment with $l = 15$ and $m = 2$ (Cubuk et al., 2020). We also use linear learning rate warmup until 10k steps and linearly decay it until the end. Our model is pre-trained with 224x224-sized images.

| Model | Dataset | Epochs | Base $\eta$ | TPU v4 hours |
|---|---|---|---|---|
| ViT-B/16 | ImageNet-21k | 300 | $10^{-3}$ | 2.7k |
| ViT-B/16 | Imagenet-1K | 300 | $10^{-3}$ | 0.4k |

Table 5: Pre-training hyperparams. We used batch size of 4096, learning rate warmup of 10k steps and then linear decay. Additionally, we set dropout rate to 0.0, clip global norm to 1 and weight decay to 0.0001. We use images of resolution 224x224. Note that we intentionally keep the model size the same to illustrate the effect of larger pre-training dataset and its effect on private fine-tuning.

# D Finetuning Details

## D.1 Datasets

**ImageNet-1k** We fine-tune on ImageNet train split and present the Top-1 accuracies we obtain from the official test split. Following Mehta et al. (2022), we used images of input resolution 256x256 which is central cropped from a resolution of 384x384. Note that this is slightly lower resolution and without Inception Crop (Szegedy et al., 2015) which is typically done in non-private setting. Finally, for training with DP, we fixed $\delta$ to be 8e-7.

**CIFAR-10 and CIFAR-100** Similar to above, we fine-tune on train split and present the Top-1 accuracies we obtain from the official test split. We also changed the input resolution to 256x256 which is central cropped from an image of resolution 384x384. Again, this may look a little unusual at first for CIFAR-10 and

CIFAR-100 since the original resolution of the images is 32x32. But we first upsample them to 384x384 and then central crop them. We found that using higher resolution images made a big difference in performance (even in non-private setting), especially when using features from a pre-trained model. Finally, for training with DP, we fixed $\delta$ to be 1e-5.

## D.2 Compute requirement

To illustrate that our proposed methods don't impose significant compute overhead, we also report the exact compute required for each of our methods in Table 6 with ImageNet-1k features (largest dataset we consider) obtained from ViT-G/14 pretrained with JFT-3B dataset (highest dimensional model we consider).

As shown in Table 6, not only the absolute compute requirement for private finetuning using our methods is quite minimal, our best performing method (DP-FC) requires only slightly higher compute overhead over our baseline (DP-Adam) while outperforming in terms of quality significantly.

| Pretraining Dataset | Method | Epochs | TPUv4 hours |
|---|---|---|---|
| | DP-LS | 1 | 0.9 |
| | DP-Newton | 1 | 0.5 |
| | | 10 | 2.4 |
| JFT | DP-FC | 1 | 0.4 |
| | | 10 | 1.2 |
| | DP-Adam | 1 | 0.2 |
| | | 10 | 0.9 |
| | | 100 | 8.7 |

Table 6: Comparison of TPU v4 hours required as a measure of exact compute required for private finetuning with ImageNet-1k features obtained from ViT-G/14 pretrained with JFT-3B dataset.

## D.3 Hyperparameter Tuning

All of our results consider full-batch setting with a constant learning rate (when applicable). Additionally, following Mehta et al. (2022), we set initial weights to 0.0.

To gain confidence in our proposed methods, we wanted to properly tune the important hyperparameters. Fortuitously, since we are only interested in learning just the last layer from features, each training run can be relatively inexpensive. To properly tune hyperparameters, we employed a Bayesian optimization package called Vizier (Golovin et al., 2017; Song et al., 2022). For tuning hyperparameters, we heldout 5% of the training set as our validation set and report top-1 accuracies on on the test set by using the tuned hyperparameters.

Note that, for a fair comparison, similar to previous work and our baseline (DP-SGD), we do not account for the privacy cost of hyperparameter tuning in our results. For proper tuning and trustful results, we made sure to run more trails than required for tuning each method, although for all of tuning runs, however we found that the optimizer converged quite early on, suggesting that the cost of hyperparamter tuning can be further reduced.

### D.4   DP-Adam hyperparameters

| Model | Pre-training DS | Fine-tuning DS | $\eta$ | $\lambda$ | DP Clipping Norm ($C$) |
|---|---|---|---|---|---|
| ViT-G/14 | JFT-3B | ImageNet-1k | $[10^{-8}, 10^8]$ | $[10^{-8}, 10^8]$ | 1.0 |
| ViT-G/14 | JFT-3B | CIFAR-10 | $[10^{-8}, 10^8]$ | $[10^{-8}, 10^8]$ | 0.005 |
| ViT-G/14 | JFT-3B | CIFAR-100 | $[10^{-8}, 10^8]$ | $[10^{-8}, 10^8]$ | 0.005 |
| ViT-B/16 | ImageNet-21k | ImageNet-1k | $[10^{-8}, 10^8]$ | $[10^{-8}, 10^8]$ | 0.005 |
| ViT-B/16 | ImageNet-21k | CIFAR-10 | $[10^{-8}, 10^8]$ | $[10^{-8}, 10^8]$ | 0.005 |
| ViT-B/16 | ImageNet-21k | CIFAR-100 | $[10^{-8}, 10^8]$ | $[10^{-8}, 10^8]$ | 0.005 |
| ViT-B/16 | ImageNet-1k | CIFAR-10 | $[10^{-8}, 10^8]$ | $[10^{-8}, 10^8]$ | 0.005 |
| ViT-B/16 | ImageNet-1k | CIFAR-100 | $[10^{-8}, 10^8]$ | $[10^{-8}, 10^8]$ | 0.005 |

Table 7: Fine-tuning hyperparams for DP-Adam. All models are trained in full-batch setting with a constant learning rate and no dropout. When training the models with DP, we replace the global clipping with per example clipping norm as specified in the table. Following Mehta et al. (2022), we set initial weights to 0.0, bias to -10.0 and train with sigmoid cross-entropy loss. Note that we employed a Bayesian optimization package called Vizier (Golovin et al., 2017; Song et al., 2022) and used a total of 200 trials for jointly tuning both the learning rate and weight decay as specified in the table.

### D.5   DP-LS hyperparameters

| Model | Pre-training DS | Fine-tuning DS | $\alpha$ | $\lambda$ | DP Clipping Norm ($C$) |
|---|---|---|---|---|---|
| ViT-G/14 | JFT-3B | ImageNet-1k | $[10^{-8}, 10^8]$ | $[10^{-8}, 10^8]$ | 1.0 |
| ViT-G/14 | JFT-3B | CIFAR-10 | $[10^{-8}, 10^8]$ | $[10^{-8}, 10^8]$ | 0.005 |
| ViT-G/14 | JFT-3B | CIFAR-100 | $[10^{-8}, 10^8]$ | $[10^{-8}, 10^8]$ | 0.005 |
| ViT-B/16 | ImageNet-21k | ImageNet-1k | $[10^{-8}, 10^8]$ | $[10^{-8}, 10^8]$ | 0.005 |
| ViT-B/16 | ImageNet-21k | CIFAR-10 | $[10^{-8}, 10^8]$ | $[10^{-8}, 10^8]$ | 0.005 |
| ViT-B/16 | ImageNet-21k | CIFAR-100 | $[10^{-8}, 10^8]$ | $[10^{-8}, 10^8]$ | 0.005 |
| ViT-B/16 | ImageNet-1k | CIFAR-10 | $[10^{-8}, 10^8]$ | $[10^{-8}, 10^8]$ | 0.005 |
| ViT-B/16 | ImageNet-1k | CIFAR-100 | $[10^{-8}, 10^8]$ | $[10^{-8}, 10^8]$ | 0.005 |

Table 8: Fine-tuning hyperparams for DP-LS. All models are trained in full-batch setting. When training the models with DP, we replace the global clipping with per example clipping norm as specified in the table. Specific to DP-LS, we clip the RHS with $C$ and the gramians with $C^2$. Interestingly, least squares is invariant to the starting weights which takes an important confounding factor away from the private training procedure. Similar to DP-Adam, we employed a Bayesian optimization package called Vizier (Golovin et al., 2017; Song et al., 2022) and used a total of 200 trials for jointly tuning both the $\alpha$ and weight decay as specified in the table.

### D.6 DP-Newton hyperparameters

| Model | Pre-training DS | Fine-tuning DS | $\eta$ | $\lambda$ | DP Clipping Norm ($C$) |
|---|---|---|---|---|---|
| ViT-G/14 | JFT-3B | ImageNet-1k | $[10^{-8}, 10^8]$ | $[10^{-8}, 10^8]$ | 1.0 |
| ViT-G/14 | JFT-3B | CIFAR-10 | $[10^{-8}, 10^8]$ | $[10^{-8}, 10^8]$ | 0.005 |
| ViT-G/14 | JFT-3B | CIFAR-100 | $[10^{-8}, 10^8]$ | $[10^{-8}, 10^8]$ | 0.005 |
| ViT-B/16 | ImageNet-21k | ImageNet-1k | $[10^{-8}, 10^8]$ | $[10^{-8}, 10^8]$ | 0.005 |
| ViT-B/16 | ImageNet-21k | CIFAR-10 | $[10^{-8}, 10^8]$ | $[10^{-8}, 10^8]$ | 0.005 |
| ViT-B/16 | ImageNet-21k | CIFAR-100 | $[10^{-8}, 10^8]$ | $[10^{-8}, 10^8]$ | 0.005 |
| ViT-B/16 | ImageNet-1k | CIFAR-10 | $[10^{-8}, 10^8]$ | $[10^{-8}, 10^8]$ | 0.005 |
| ViT-B/16 | ImageNet-1k | CIFAR-100 | $[10^{-8}, 10^8]$ | $[10^{-8}, 10^8]$ | 0.005 |

Table 9: Fine-tuning hyperparams for DP-Newton. All models are trained in full-batch setting with a constant learning rate and no dropout. When training the models with DP, we replace the global clipping with per example clipping norm as specified in the table. Specific to DP-Newton, we clip the features instead of the gradient and the Hessian. We do this save on computational cost since now we don't need to explicitly compute per-example gradient and the Hessian. For the privacy analysis, we use the clipping norm $C$ to sanitize the gradient and $\frac{C^2}{4.0}$ for the Hessian. Following Mehta et al. (2022), we set initial weights to 0.0 and train with sigmoid cross-entropy loss. We employed a Bayesian optimization package called Vizier (Golovin et al., 2017; Song et al., 2022) and used a total of 300 trials for jointly tuning both the learning rate and weight decay as specified in the table.

### D.7 DP-FC hyperparameters

| Model | Pretraining DS | Finetuning DS | $\eta$ | $\lambda$ | DP Clipping Norm ($C$) |
|---|---|---|---|---|---|
| ViT-G/14 | JFT-3B | ImageNet-1k | $[10^{-8}, 10^8]$ | $[10^{-8}, 10^8]$ | 1.0 |
| ViT-G/14 | JFT-3B | CIFAR-10 | $[10^{-8}, 10^8]$ | $[10^{-8}, 10^8]$ | 0.005 |
| ViT-G/14 | JFT-3B | CIFAR-100 | $[10^{-8}, 10^8]$ | $[10^{-8}, 10^8]$ | 0.005 |
| ViT-B/16 | ImageNet-21k | ImageNet-1k | $[10^{-8}, 10^8]$ | $[10^{-8}, 10^8]$ | 0.005 |
| ViT-B/16 | ImageNet-21k | CIFAR-10 | $[10^{-8}, 10^8]$ | $[10^{-8}, 10^8]$ | 0.005 |
| ViT-B/16 | ImageNet-21k | CIFAR-100 | $[10^{-8}, 10^8]$ | $[10^{-8}, 10^8]$ | 0.005 |
| ViT-B/16 | ImageNet-1k | CIFAR-10 | $[10^{-8}, 10^8]$ | $[10^{-8}, 10^8]$ | 0.005 |
| ViT-B/16 | ImageNet-1k | CIFAR-100 | $[10^{-8}, 10^8]$ | $[10^{-8}, 10^8]$ | 0.005 |

Table 10: Fine-tuning hyperparams for DP-FC. All models are trained in full-batch setting with a constant learning rate and no dropout. When training the models with DP, we replace the global clipping with per example clipping norm as specified in the table. Specific to DP-FC, we clip the per-example gradients with clipping norm $C$ and $C^2$ for the gramian which is shared across classes and iterations. Following Mehta et al. (2022), we set initial weights to 0.0, set initial bias to -10.0 and train with sigmoid cross-entropy loss. We employed a Bayesian optimization package called Vizier (Golovin et al., 2017; Song et al., 2022) and used a total of 200 trials for jointly tuning both the learning rate and weight decay as specified in the table.

