# OpenReview forum: "Differentially Private Image Classification from Features"
_TMLR — Accepted by TMLR_

### Review · Reviewer_nzZQ · 2022-12-12

**Summary Of Contributions:**

This paper studies transfer learning for image classification models under differential privacy. The underlying task is the typical task in DP transfer learning, where a complex classification model is pre-trained using a publicly available data set, and then the model is fine-tuned for a sensitive private data with DP optimization methods. The previous approaches in the topic have used DP-SGD for the fine-tuning part.

In this paper, authors suggest three new algorithms for the private fine-tuning part with focus on large scale image classification models. First of the methods, the DP-Newton (Alg. 2) is a simple extension of DP-SGD, which uses the perturbed Hessian as well as the perturbed gradients. The main benefit of the method is to use the second order information for the optimization. The second method proposed, the DP least squares (DP-LS, Alg. 3), uses a least squares loss instead of the logistic loss to optimize the weights. Essentially, the model is a linear regression instead of logistic regression. For this approach, authors can write the optimal parameters in terms of sufficient statistics and, perturbing those achieve DP. The 2nd order information, or the Hessian, arises naturally in the sufficient statistics for this model. The final method, the DP feature covariance (DP-FC, Alg. 4), reverts back to the logistic loss, but uses the perturbed empirical feature covariance as a preconditioned for the gradients. The empirical covariance matrix is treated as an approximation of the 2nd order information, and compared to the DP-Newton, it only needs to be computed once, thus saving in DP budget.

The empirical performance of the methods seems really impressive, with DP-FC providing significant improvements to the SOTA (see Table 1).

**Audience:**

Yes

**Broader Impact Concerns:**

I don't think the work needs a broader impact statement.

**Claims And Evidence:**

No

**Requested Changes:**

I want authors to clarify the following points I raised in the weaknesses:
- Is there a bound for the linear regression loss function's derivative
- Clarification for the hyperparameter tuning setup
- Some error bounds for the methods (more repeats)

Additionally, making the code available would help a lot.

**Strengths And Weaknesses:**

## Strengths
The proposed methods seem like intuitive extensions to the standard DP-SGD, and under the limited number of gradient updates (<=100, with full batch in the experiments) the 2nd order information might indeed provide significant improvement to the learning (at least it would in non-DP case). The empirical result seem to indeed support this, with the higher order methods constantly outperforming the DP-SGD.

## Weaknesses
However, I do have some concerns which I hope the authors can address.

### The bound for the linear regression loss
In the paragraph after Thm 2, you state that $|\ell'(z,y)| \leq 1$ holds for the squared loss for _all_ $(z,y)$. How can this be? You have $\ell'(z,y)=z-y$, and now if your $z>1$ the claim obviously does not hold for $y=0$. I don't think there is a bound for the $z$ even if your input $x$ would be bounded as the regression parameters will be in $\mathbb{R}^d$. If this does not hold, then I'm not sure if the reported $\epsilon$ values for the DP-LS are correct.

### Hyperparameter tuning
The appendix seems to suggest that you have used Bayes optimization with 200 trials in choosing the hyperparameters for you fine-tuning. If I'm correct, the privacy cost of this is not accounted in the reported $\epsilon$ values (by the way: please state this explicitly). Now, in general I don't find the non-DP hyperparameter tuning such a big issue, as that is what many works in the field seem to do. However, for the particular method proposed, the hyperparameter tuning seems like a hugely important thing. For example, if you consider the single epoch, full batch (i.e. single step) gradient optimization, the hyperparameter search starts to resemble line search, which I would image has a huge effect on the performance. Therefore, if the results are really strongly dependent on the chosen hyperparameters, authors should definitely be more up-front about it.

Also, I would like the authors to better explain the hyperparameter tuning setting. Is there a separate validation set on which you tune the hyperparameters? Or are you just using the train split for the hyperparameter tuning?

### Robustness of the results
All the tables report single top-1 accuracy results. This leaves me to wonder how robust the results actually are. For example, does the DP-Newton give you constantly top-1 accuracy of ~70% for the epsilon=0.01 setting of Table 2? What would happen if you would repeat the inference for example 10 times? Please, do add some error bounds for the methods. Otherwise it is really impossible to estimate the consistency of the methods.

## Other comments/suggestions/questions
- I believe the preconditioner used in Alg. 4 is a scaled version of the Fisher information matrix of the logistic regression log-probability. The Hessian for this particular problem will be block diagonal, and the $\nabla^2 log p(y \mid x, \theta)$ is given as $\sigma(\langle\theta, x\rangle)(1-\sigma(\langle\theta, x\rangle)) x x^T$. All the blocks of the Fisher information will therefore be scaled with a scalar. Therefore, I think the learning procedure of Alg. 4 with the empirical covariance matrix as preconditioner resembles the one of natural gradients, and could be maybe motivated through it. It would be really interesting to see if adding the correct scaling to the preconditioner would yield even better results for the non-private classifier (this is just a curiosity though).

- I think it would be fair if the Tables would include the best non-DP accuracy as well. Now it remains unclear how much the epoch limitation for example affects the non-DP baseline. I know that this is not probably attainable baseline, but it would give the reader relevant information of how well we should expect these algorithms to perform in non-DP case.

- I guess in the evaluation of the linear regression based solution you simply pick the category $j$ with highest $\langle \theta_j, x_i\rangle$?

- I must say, that I'm bit puzzled by some results. For example, in Table 2, the results of a single step DP-Adam with $\epsilon=0.01$ vs the DP-Newton in the same setting are really surprising. The only difference between these methods, as far as I understand, is the Hessian, and the gradients in DP-Adam should only have half the noise due to budget splitting in DP-Newton. I would imagine that if we do only a single step of the learning, the noisy (really noisy in this case) Hessian would not help the learning that much, but it seems that the difference is really drastic with DP-Newton giving top-1 accuracy of ~70% while DP-Adam <20%! Just to be clear, I'm not claiming that the results are wrong, but I would like to understand better how does this happen. Also, *making your source code available* would make it easier to understand the algorithms and the parameter choises used.

### Typos and other minor things:
- "... only with the number of positive classes per example": To me this would make more sense if it would be "positive examples per class", but maybe I misunderstood something. Or can an example (which I think is the sample?) have multiple different classes (which I believe are the labels)?
- In the Appendix B.1 you state "satisfies a much stronger property of zCDP ≤ 1 (0.154 for ε = 4) which is by now an industry standard.". Is this because the conversion from zCDP to approximate DP is lossy? I'm not too familiar with the industry standards in terms of zCDP, so any citation that zCDP $\leq 1$ is used in practice would be helpful.
- It seems that in Algs 1 and 4 you have used $\ell: \mathbb{R}^{m \times d}$ and in Alg 2 just $\ell: \mathbb{R}^{m \times d}$. I think the latter one must be a typo right?
- Are you considering the add/remove neighborhood relation for the privacy analysis? I guess so, as otherwise the sensitivities should have a $\sqrt{2}$ before them. Might be good to mention it, now you have just stated that $D$ and $D'$ differ in a single data point, which could also refer to the substitute neighborhood.

---

> ### Author Response · Authors · 2023-02-02
> **Reply to reviewer nzZQ**
>
> We thank the reviewer for the detailed and thoughtful review! We hope that our following response alleviates some of your concerns:
>
> ``The bound for the linear regression loss``
>
> If you're referring to the statement below Theorem 2.1, then you're absolutely correct i.e. linear regression loss gradient would not satisfy $|\ell'(z,y)| \leq 1$ for all $(z, y)$. But because for a quadratic loss, Newton's method reaches the optimum in a single step (no matter where it is initialized), we always initialize it at 0 and run it for a single step (which is why the number of “epochs” is 1 in Table 2 - notice that the loop in Algorithm 3 is over classes, not iterations). Thus, our algorithm requires only computation of the gradient and Hessian at the *origin* (i.e. $z=0$). At the origin we of course have $\ell’(z,y) = -y$, which is bounded since the labels $y$ are bounded. Because of this, we are able to bound the loss gradient and Hessians at the origin and so achieve DP for linear regression.
>
> We will amend the discussion after Theorem 2.1, to clarify that for quadratic loss, we initialize the algorithm at the origin, and run it for a single step.
>
>
> ``Hyperparameter tuning``
>
> You are right that the privacy cost of hyper-parameter tuning was intentionally not budgeted in our privacy accounting. As you pointed out, this is in line with the setup of the baselines we consider and allows for a fair comparison with reported numbers. However, we totally agree with your concern. We have in fact updated our manuscript to that effect and we explicitly mention this even in our "Limitations" section.
>
> We would additionally like to note that, in this work, we resort to more extensive hyperparam tuning neither because it is necessarily needed nor that methods are sensitive to it, but we do it because these methods are quite new and we wanted to have a well-tuned set of results that we can trust, and can derive meaningful conclusions from. In fact, we have found that results in for instance, single epoch, full batch setting are actually quite robust to the exact value of the learning rate (most important hyperparam) as long as it is low enough to not diverge.
>
> We also agree with your point about including more information on our hyperparameter tuning! It is indeed valuable to put in the manuscript and we do so in Appendix Section D.3. To specifically answer your question, we did hold out 5% of the training data for hyperparameter tuning for all the fine tuning datasets.
>
> ``Robustness of the results``
>
> Again, you are absolutely right! Based on your feedback, we have updated our main set of results i.e. Table 1 to include both the median and stddev from 5 runs with different seeds. We didn't realize at first but your intuition is spot on that stddev is higher for stricter privacy budgets (depending on the method and dataset used).
>
> ``I believe the preconditioner used in Alg. 4 is a scaled version of the Fisher information matrix``
>
> You are of course correct that there is a close connection between second-order methods in logistic regression with a linear model and the natural gradient algorithm. We had not considered this viewpoint in our work, but we agree that by tweaking the constant preconditioner of Algorithm 4 as you suggest, we may be able to produce an algorithm that is even more theoretically motivated and hopefully also better in practice. This is a great idea that we hope to incorporate into future experiments!
>
> ``I think it would be fair if the Tables would include the best non-DP accuracy as well.``
>
> You are right. We have included the best non-DP accuracy in the captions now.
>
> Additionally, we would like to note that, we are not necessarily arbitrarily restricting our methods to be epoch-limited. We do this to save on the privacy budget. This is best illustrated by looking at DP-Adam performance where as we go from 1 epoch to 10 epochs improves performance but the performance degrades when going to 100 epochs where the noise from DP overpowers any improvements coming from extra compute.
>
> **Continued to next comment** ==>

---

> > ### Author Response · Authors · 2023-02-02
> > **Reply continued**
> >
> > ``I guess in the evaluation of the linear regression based solution you simply pick the category``
> >
> > Yes, exactly.
> >
> > ``I must say, that I'm bit puzzled by some results..``
> >
> > Thanks for pointing this out! While we do not have a complete explanation of why this happens, we have empirically observed that 2nd order information almost always helps in our full batch setting, even in the single step scenario. Performance typically improves when we go from DP-SGD to DP-Adam (also observed by https://arxiv.org/abs/2205.02973) and further improves from DP-Adam to DP-Newton.
> >
> > Regarding code: absolutely! We are working hard on being able to release our code. We will open-source our code as part of this publication to garner feedback from the research community.
> >
> > ``... only with the number of positive classes per example``
> >
> > Apologies for any confusion. The formulation of DP-LS indeed allows for multiple positive labels (or classes) per example. Although, the datasets we use only consider a single class setting i.e. only 1 positive class per example.
> >
> > ``In the Appendix B.1 you state "satisfies a much stronger property of zCDP..``
> >
> > Definitely! We relied on the following industry articles to justify this statement:
> >
> > 1. Google AI: https://ai.googleblog.com/2022/02/federated-learning-with-formal.html
> > 2. US Census: https://www2.census.gov/about/training-workshops/2021/2021-07-01-das-presentation.pdf
> >
> >
> > ``Are you considering the add/remove neighborhood relation for the privacy analysis?``
> >
> > Yes exactly! Apologies for any confusion.

---

### Review · Reviewer_Diqz · 2022-12-21

**Summary Of Contributions:**

This paper tries to improve the utility of differentially private learning by reforming the loss function and optimization approach. Results show somewhat improvement compared with previous work.

**Audience:**

Yes

**Claims And Evidence:**

Yes

**Requested Changes:**

Please address my previous concerns

**Strengths And Weaknesses:**

The aim of this paper is to improve the utility of differentially private learning. It finds that by changing the loss function (from logistic regression to least-square linear regression), the performance is improved, moreover, by using second-order optimization methods, the performance can also be improved. This paper combines such two techs. to design the DP-FC algorithm which shows better results compared with previous SOTAs.
However, the advantage is not significant as can be seen from Table 1 if compared with previous SOTAs. The cost of the proposed algorithm which is the computational cost (line 2 of alg. 4) actually offsets the advantage.
Overall, the pre-training paradigm is not new, DP-FC offers limited improvements while also incurring quadratic computation costs with respect to the number of samples.  The proposed method may not constitute a satisfying manuscript for potential publication.

---

> ### Author Response · Authors · 2023-02-02
> **Reply to reviewer Diqz**
>
> We thank the reviewer for the review! Please see our reply below to address the specific concerns related to cost and our general view at the end. We hope that our response is helpful and you would consider supporting our paper! Definitely let us know if you have any other comments/questions.
>
> ``The cost of the proposed algorithm which is the computational cost (line 2 of alg. 4) actually offsets the advantage.``
>
> It looks like the reviewer is concerned about a particular step in DP-FC algorithm where we compute the feature covariance ($G = \sum_{i=1}^n (x_i x_i^\top)$). This step is O(n*d^2) where n=number of samples and d=feature dimension. We would like to note that this is in fact a relatively small computation! Indeed, in our experiments, the wall clock time of training with DP-FC is roughly the same as training with DP-Adam (keeping everything else constant i.e. number of devices, epochs, batch size etc).
>
> Moreover, we compute the sum of the outer product of the input features once and cache the sanitized version for reuse throughout the steps of optimization to reduce the cost even further.
>
> Maybe we are misunderstanding the reviewer's concern, in which case, please let us know!
>
> ``Overall, the pre-training paradigm is not new, DP-FC offers limited improvements while also incurring quadratic computation costs with respect to the number of samples..``
>
> Your calculation correctly observes that the complexity is quadratic. However, as mentioned above, you may have overlooked that the the complexity is actually quadratic w.r.t the **feature dimension** and not the number of samples.
>
> We agree that a quadratic complexity w.r.t number of samples, while still interesting in theory, indeed would have made our approach much less appealing in practice. But fortunately, that is not the case!
>
> ``Our general view on computational cost with DP``
>
> While computational cost is a very important concern, we would like to note that training with DP has fundamentally different tradeoffs when compared to traditional deep learning training.
>
> In deep learning, without any DP constraints, one can almost deterministically improve performance by increasing compute (and/or memory) by either adding more parameters to the model or by training longer. We observe that this stops being true with DP constraints since increasing both dimensionality and epoch count increases the amount of noise and ultimately degrades performance after a point.
>
> Thus, a method with significantly increased costs may still be interesting (and non-trivial) if it can improve performance while keeping the privacy budget constant -- although our method is *not* one with significantly increased costs.

---

### Review · Reviewer_db1x · 2023-01-30

**Summary Of Contributions:**

This paper focuses specifically on the image classification and studies the last-layer fine-tuning. The authors propose three new second-order methods, DP-Newton, DP-FC, DP-LS, to fine-tune Vision Transformers on 3 datasets. The main contributions are on the methodology, which come with formal privacy guarantee, and on the image experiments. Through their analysis, we may advance our understanding of pretraining datasets and optimizers from an empirical perspective.

**Audience:**

Yes

**Claims And Evidence:**

No

**Requested Changes:**

1. The tables are hard to follow when I tried to understand which model gives which results. I searched for a while in the appendix to find this important information (because larger models generally give better results, which must not be confused with the benefits of new methods). The models should be added to Table 1-4 as an additional column.

2. Unfair comparison is observed in the tables. An experiment is of the following elements: (pretraining dataset, optimizer, training time/computation resource, model, fine-tuning method). Given that the main contribution of this paper is on the `optimizer' part, other parts should be set similar to previous results.

In Table 1, previous SOTA on CIFAR10 and CIFAR100 are pretrained on ImageNet21k and trained for 3 epochs, while the authors achieve new SOTA on a LARGER pretraining dataset JFT, MORE epochs (10 epochs) and LARGER model (vit-G 2B v.s. BeiT-large 0.3B). This is for sure not an apple-to-apple comparison because the model is different, the training cost is different, and the pretraining is different. I agree that it is not necessary to use the same setting as previous SOTA. However, the more different your setting is, the less convincing that your new optimizer is better. At least one experiment on CIFAR10/100 with the same pretraining dataset ImageNet21k is needed.

3. Why in Table 2/3/4, DP-LS is only trained for 1 epoch, unlike DP-FC and DP-Newton which also show the 10 epoch results? The 1 epoch result shows that DP-LS is often the best method, so it may be true that it is also the best if trained for 10 epochs. Also DP-LS is strong when epsilon is large (e.g. Table 2, epsilon=8, DP-LS 1 epoch=86.7, DP-Newton=84.9, DP-FC=85.6), this contradicts your claim in the abstract "least squares is more effective at lower epsilon values while Newton’s method is more effective at larger epsilon values".

3. Discuss efficiency. I would at least need the training time of each method. In the paper, most discussion is about the number of epochs, which is unimportant if DP-FC takes much longer to train one epoch than DP-SGD/Adam. This renders the claim in Section 3.2 "(DP-FC) allow us to obtain better results with 10 epochs compared to even when using 100 epochs with DP-SGD" not well-supported.

**Strengths And Weaknesses:**

Strengths:
The main strength is the novelty in the methodology, especially the DP-FC, which is well-explained by firstly define and experiment on DP-LS and DP-Newton. The paper is easy to follow and solid in terms of DP guarantee. In Section 2.3, DP-LS is carefully described and distinguished from existing work that establish on the closed-form solution of least square problems (maybe a missing reference to https://arxiv.org/abs/1803.02596). The experiments are convincing and supporting the claims well.

Weaknesses:
While this paper is well-written and fulfills its scope, the contributions are mostly empirical and on the privacy theory. From an algorithmic viewpoint, an extended discussion on the efficiency, the convergence theory, and the applicability will enhance the paper significantly.

I understand that the efficiency is briefly discussed in Section 4, but no time/space complexity or empirical evidence such as training time per epoch and RAM occupation is present. This could be very meaningful to have given that second-order information is used, whereas most existing DP works use first-order ones.

Besides, although this paper makes it clear that the focus is on the image classification, it is desirable to at least discuss whether this method works on language models and tasks. On a side note, it is commonly observed that last-layer fine-tuning does not work well with language models. Therefore, if DP-FC generally works on language models, it would be a great advantage.

Also, I highly recommend not to use JFT. JFT is a proprietary dataset that is not public. Any experiment on it is not reproducible if not hard-to-trust, it is not a good practice for the DP community. See more comments in 'Requested Changes'.

---

> ### Author Response · Authors · 2023-02-02
> **Reply to reviewer db1x**
>
> We thank the reviewer for kind words, the review and suggestions! Please see our reply below to address the specific concerns.
>
> ``Main concern regarding efficiency``
>
> You are absolutely right that more information on efficiency would make the paper stronger! We have added more discussion on it and exact running times of different methods in Appendix Section D.2. Specifically, since the computation of feature covariance in DP-FC is the only additional cost compared to DP-Adam (which is further reduced by caching feature covariance across training steps), we find that the compute requirements for both are not very different on actual hardware.
>
> More generally though, while computational cost is a very important concern, we would like to note that training with DP has fundamentally different tradeoffs when compared to traditional deep learning training. Without any DP constraints, in deep learning, one can almost deterministically improve performance by increasing compute (and/or memory) by either adding more parameters to the model or by training longer. We observe that this stops being true with DP constraints since increasing both dimensionality and epoch count increases the amount of noise and ultimately degrades performance after a point.
>
> Thus, we would like to argue that a method with significantly increased costs may still be interesting (and non-trivial) if it can improve performance while keeping the privacy budget constant - although our method is *not* one with significantly increased costs.
>
> ``Use of JFT``
>
> You're right that JFT is a proprietary dataset and it's hard to reproduce results derived from it. We have indeed cited this as a limitation of this work in Appendix Section A.
>
> Having said that, JFT is a dataset used by some of the most impactful works outside of DP. In addition, it lets us do a fair comparison with other relevant works which do include results with JFT, for instance: https://arxiv.org/abs/2205.02973 and https://arxiv.org/abs/2204.13650. Finally, we would like to note that, in order to make our work more generalizable and reproducible, we also include results with models pre-trained with ImageNet-21k and ImageNet-1k. Our claims and conclusions are derived from results obtained with all 3 pre-training datasets in aggregate. But, definitely let us know if you had any specific issues, we would be happy to address them.
>
>
> ``Missing reference``
>
> Thank you for suggesting the reference! Indeed, we are also admirers of AdaSSP and other contributions in that paper; we have added that as a citation in our revision.
>
> **Continued to next comment** ==>

---

> > ### Author Response · Authors · 2023-02-02
> > **Reply continued**
> >
> > ``Discussion on tasks other than image classification``
> >
> > You are absolutely right! We have added this as a limitation of this work and definitely agree that it is indeed a valuable direction.
> >
> > ``Model information in tables``
> >
> > We agree with you. Although we do provide this information in Appendix Section C.2, it would indeed be useful to put it close to the results. Since we broke down our results with one pre-training model/dataset per table (except for Table 1), we have now included model information in the caption with each table. We hope that this is useful!
> >
> > ``Unfair comparison is observed in the tables``
> >
> > We assume you are talking about just Table 1 since that is the only set of results where we compare with the results reported by previous works. Except for Table 1, we indeed fix everything else except for the optimizer, like you mentioned (with DP-Adam as our baseline). Our use of strong pre-trained features stems from well established knowledge in the community effectiveness of transfer learning for DP.
> >
> > For Table 1, our intention was to compile our best results for each dataset and provide another interesting axis to view our results from. We additionally include SOTA numbers from previous work just as a reference. We indeed don't intend to claim that the usage of our proposed methods alone would lead to these results. Definitely let us know if we inadvertently over-claimed in our manuscript, we would be happy to correct it!
> >
> > ``Why in Table 2/3/4, DP-LS is only trained for 1 epoch..``
> >
> > Interesting question! Training DP-LS for more epochs would definitely be quite appealing especially given that it performs so well when trained for 1 epoch. But notice that, DP-LS has an exact solution (just below equation 6) which does not depend on the current parameters ($\theta$) at all. This means that even if we were to take more steps, the solution would not change (barring the noise coming from DP). Now since DP noise increases with epoch count, epoch=1, in fact, is the best one can potentially do with DP-LS.
> >
> > Regarding comparing DP-LS with epoch=1 and DP-FC with epoch=1:  as we mentioned earlier, DP considerably changes the relationship between end performance and compute. With DP constraints, it is no longer necessarily possible to improve performance just by training longer. Thus, we actually intend to compare our methods in an unrestricted compute (or open-compute) setting. So, when we are comparing 2 methods, we are comparing the best performance obtained by each method across the number of epochs we considered for them. Having said that, while all of our conclusions/claims are derived in an open-compute setting, our results also would also allow us to make conclusions in a compute-restricted setting. For instance, as you pointed out, if we fix the number of epochs=1, DP-LS is indeed a very strong method across the board!

---

### Author Response · Authors · 2023-03-06
**General comment**

We thank the reviewers for taking the time to review our submission and for great feedback! We especially appreciate that they truly engaged with our submission and critically assessed the work. Definitely let us know if there were any additional concerns we can address, we would be more than happy to do so!

-Authors

---

### Decision · Action_Editors · 2023-03-25

**Recommendation:** Accept with minor revision

**Comment:**

The paper makes contributions to transfer learning of image classification under differential privacy. There are threoretical results proved for the three new proposed methods for differentially privately optimizing the fine-tuning aspect of training (model is trained non-privately on public data, and then only fine-tuned on sensitive data for which privacy concerns are important). There is substantial empirical evaluation that shows the efficacy of the proposed methods.

The reviewers have made several useful comments which the authors should incorporate into the main article. In particular, the authors should incorporate all the changes that they have agreed to make in response to the reviews.

They should also be explicity about their intentions to release code. One reviewer has rightly suggested that in order for this paper to make most impact, and for reproducibility reasons, code release is highly desirable.

**Audience:**

Primarily researchers with interest in theory and practical application of Differential Privacy, but would also be of potential interest to researchers who are more broadly interested in optimization, generalization and robustness of Deep Neural Nets.

**Claims And Evidence:**

Yes, the claims in the paper are adequately supported. There are some interesting theoretical results as well as substantial empirical evaluation.

---

> ### Author Response · Authors · 2023-04-07
> **Thank you**
>
> We would like to thank the editors and the reviewers! Your feedback has been quite valuable and has made the paper stronger. We are glad to see our work accepted at TMLR.
>
>
> ```Regarding incorporating changes mentioned in comments```
>
> For sure! we had already incorporated all the useful changes suggested by the reviewers. Definitely let us know in case we missed any.
>
> ```Regarding the code```
>
> Absolutely! In our camera-ready version, we have included a link to the code we had already open-sourced. We welcome your feedback and suggestions, if any.